# Application of Peptides in Construction of Nonviral Vectors for Gene Delivery

**DOI:** 10.3390/nano12224076

**Published:** 2022-11-19

**Authors:** Yujie Yang, Zhen Liu, Hongchao Ma, Meiwen Cao

**Affiliations:** State Key Laboratory of Heavy Oil Processing, Department of Biological and Energy Chemical Engineering, College of Chemical Engineering, China University of Petroleum (East China), 66 Changjiang West Road, Qingdao 266580, China

**Keywords:** peptide, gene delivery, nonviral vector, self-assembly, gene therapy

## Abstract

Gene therapy, which aims to cure diseases by knocking out, editing, correcting or compensating abnormal genes, provides new strategies for the treatment of tumors, genetic diseases and other diseases that are closely related to human gene abnormalities. In order to deliver genes efficiently to abnormal sites in vivo to achieve therapeutic effects, a variety of gene vectors have been designed. Among them, peptide-based vectors show superior advantages because of their ease of design, perfect biocompatibility and safety. Rationally designed peptides can carry nucleic acids into cells to perform therapeutic effects by overcoming a series of biological barriers including cellular uptake, endosomal escape, nuclear entrance and so on. Moreover, peptides can also be incorporated into other delivery systems as functional segments. In this review, we referred to the biological barriers for gene delivery in vivo and discussed several kinds of peptide-based nonviral gene vectors developed for overcoming these barriers. These vectors can deliver different types of genetic materials into targeted cells/tissues individually or in combination by having specific structure–function relationships. Based on the general review of peptide-based gene delivery systems, the current challenges and future perspectives in development of peptidic nonviral vectors for clinical applications were also put forward, with the aim of providing guidance towards the rational design and development of such systems.

## 1. Introduction

Gene therapy refers to the introduction of exogenous genes into target cells to correct defective and abnormal genes for the purpose of treating diseases. With the development of modern molecular biology and progress of human genome project, gene therapy has become a promising strategy to treat cancer, gene diseases, infectious diseases, cardiovascular diseases and nervous system diseases [1]. The therapeutic nucleic acids used in gene therapy include plasmid DNA, siRNA and other free nucleic acids [2]. However, it is difficult for these nucleic acids to reach the target tissue due to their large molecular weight and huge number of negative charges [3]. Therefore, developing safe and effective gene delivery vectors is essential for gene therapy.

Generally, gene delivery vectors are categorized into two types, that is, viral vectors and nonviral ones. Typically, viral vectors use modified viruses including retroviruses, lentiviruses, adenoviruses and adeno-associated viruses to carry genes into cells due to their advantages of high infection level of host cells [4]. Their accurate programmed infection characteristics and efficient delivery ability of exogenous gene into host cells make them the most widely used gene vectors in clinical trials. However, viral vectors have inherent disadvantages such as potential carcinogenic effects, limited DNA encapsulation ability, lack of targeting ability and difficulty in production [5]. Moreover, they may also activate the host’s immune system and reduce the effectiveness of subsequent gene delivery [6]. These defects greatly limit the usage of virus vectors in clinical treatment and further promote the development of nonviral gene delivery systems [7]. Compared with viral vectors, nonviral vectors are usually easier to synthesize and operate, having lower immune response, larger loading capacity of genetic material and better targeting ability. Recently, a large number of efficient and safe nonviral vectors have been designed for gene therapy. When using nonviral vectors to deliver nucleic acids such as DNA [8], messenger (m)RNA [9], short interfering (si)RNA [10] and micro (mi)RNA into cells [11], they need to overcome several biological barriers (Figure 1). First, the vectors should protect the nucleic acids from degradation by endonucleases and exonucleases and help them evade immune detection [12,13,14]. Second, the vectors need to contain specific groups and ligands both to make nucleic acid molecules exude from the bloodstream to the target tissue and to mediate cell entry. Third, siRNA and miRNA mimics must be loaded into the RNA-induced silencing complex, while mRNA must bind to the translational machinery and DNA must be further transported to the nucleus to play its function (Figure 2) [15]. The commonly used nonviral vectors include cationic liposomes, cationic polymers, dendrimers, peptides and so on [16]. Among them, peptides have been considered as unique tools for delivering nucleic acid drugs due to their excellent biocompatibility and biodegradability, ease of production and modification as well as being able to respond to external stimuli [17,18,19].

Nowadays, many peptides have been incorporated as functional components into nonviral gene delivery systems to overcome various biological obstacles and deliver nucleic acid drugs to target sites with high efficiency. Peptides used as non-viral gene vectors can be divided into the following types according to their functions: cell penetrating peptides (CPPs), membrane active peptides, targeting peptides, and nuclear localization signal (NLS) peptides (Table 1). In this review, we first talk about the strategies for constructing peptide–nucleic acid complexes, and then summarize the applications of these peptides in gene delivery, as well as how to combine these peptides with other nonviral vectors to achieve the purpose of improving transfection efficiency.

## 2. Construction of Peptide–Nucleic Acid Complexes for Gene Delivery

To achieve the purpose of gene delivery, the functional peptides should be first fused with nucleic acids to form complexes so as to play the roles of gene condensing, protection, and delivery. Three main strategies can be adopted to achieve peptide/nucleic acid fusion. The first is to link the peptide segment covalently with nucleic acid to produce a conjugated molecule. For this strategy, the functional peptide segments are conjugated to the to-be-delivered nucleic acid via chemical bonds (e.g., ester bond, disulfide bridge, thiol-maleimide linkage) [73]. The superior advantage of this strategy is that the peptide–nucleic acid conjugated molecule has defined structure and stoichiometry as well as high stability, which can lead to repeatable delivery performance. This approach is particularly suitable for charge-neutral nucleic acid analogs such as phosphonodiamidate morpholino oligomer (PMO) and peptide nucleic acid (PNA) [74,75]. The peptide–nucleic acid conjugate can easily cross the cell membranes and enter the nucleus and fulfill its biological functions. Currently, this strategy has exhibited promise in clinical trials. For example, peptide-PNA conjugates have been utilized in preclinical studies targeting c-myc for severe combined immunodeficiency, while peptide-PMO conjugates have been employed for Duchenne muscular dystrophy [76,77]. However, for this strategy, the covalent bond formation may reduce the biological activity of nucleic acids or inhibit their release and expression in cells, which may hinder their application. The second is the noncovalent complexation strategy, which is to complex peptides with nucleic acids directly via noncovalent forces. For this strategy, the peptides are usually designed to have various positive charges, which can fist bind with negatively charged nucleic acids to result in charge neutralization and then induce hydrophobic collapse of the nucleic acid molecules into condensed nanoparticles [78]. This strategy has superior advantages including ease of vector construction, high loading amount of gene drug, and controllable genome release by introducing stimuli responsibility. It is suitable for delivery of most nucleic acids involving plasmid DNA, siRNA, mRNA and so on. Peptide–nucleic acid nanocomposites obtained by this method are easy to prepare and have been attempted to treat a series of diseases including cancer and cardiovascular diseases [79,80]. However, it should be noted that the peptide should be well designed to endow the peptide carrier with high functionality and avoid loss of peptide function because of its electrostatic binding with nucleic acids. The third strategy is to modify functional peptide segments on the surface of specific nanoparticles to produce composite nanoplatforms, which can further be used to complex with nucleic acids for delivery purposes. This strategy can take advantage of the nanoparticles to facilitate cellular uptake as well as to give multifunctionalities [81], which is especially suitable for development of systems for combined therapy. In summary, the above three strategies, each having specific features in peptide/nucleic acid fusion, have been extensively used in gene delivery.

## 3. Application of CPPs in Gene Delivery

Composed of 10–20 amino acids, CPPs are one class of peptides which have the potential to penetrate bio-membrane and transport bioactive substances into cells [82]. In recent years, a variety of substances such as hydrophilic proteins, nucleic acids and even nanoparticles have been carried by CPPs across cell membrane into the cytoplasm to serve specific functions. This rapid intracellular transport is not destructive to cell membranes, and the active substances can be delivered into a variety of cells regardless of the cell type. Use of CPPs to deliver nucleic acids and drugs for gene therapy and disease treatment has therefore attracted extensive attention. For example, Emma et al. designed a new 15-amino acid linear peptide CHAT that contains six arginine residues, the minimum number of residues required for cell uptake [22]. The cysteine residues located at both ends can enhance the stability of the delivery system and achieve cargo release in cells. Experiments demonstrated that CHAT peptide can transfect plasmid (p)DNA into various cell lines, resulting in successful reporter-gene expression in vivo in 4T1 and MDA-MB-231 breast xenograft models (Figure 3a). The transfection efficiency in tumor sites is comparable to that of commercial transfectants, making it a low-cost, easily formulated delivery system for the administration of nucleic acid therapeutics. However, some inherent properties of CPPs limit their clinical application. First, when CPPs are administered in vivo, they are penetrable only at concentrations above micromoles, which will cause many systemic side effects. In this case, designing new CPPs and improving their ability to penetrate cell membranes are of great importance for enhancing the safety of CPP application. Recently, a pH-active CPP called dimer LH2 was designed by Dougherty and co-workers because they found that amphiphilic CPPs in dimeric form showed higher cell-penetrating activity compared with the monomeric ones [23]. As expected, dimer LH2 can effectively deliver nucleic acid drugs to triple-negative breast cancer cell MDA-MB-231 with only tens of nanomolar concentration, showing strong membrane penetrating ability and antitumor effects [24]. In addition to using CPPs as carriers to deliver pDNA into cells, naked siRNA must be protected and delivered by carriers to enter the cell, because it is unstable, and readily degraded by nucleases in the serum environment and absorbed by tissues [83]. To solve this problem, Martina et al. used DMBT1-derived peptides with membrane penetrating ability as carriers to prepare siRNA delivery nanoparticles, which can complex with siRNA and transfect human breast metastatic adenocarcinoma MCF7 cells [25]. The delivered siRNA exhibited effective gene silencing in MCF7-recombinant cells. The study laid the foundation for developing a new vector for therapeutic siRNA delivery.

Second, most CPPs can be internalized by all cell types and lack the ability to target specific tissues as particular objectives. This imprecise feature will lead to their low stability in blood, poor tissue penetration and limited cell uptake, thus greatly reducing their targeting efficiency towards specific tissues. To solve this problem, several strategies have been developed to improve the specificity of CPPs to pathological tissues. Among them, combing targeting molecules such as RGD (Arginine-Glycine-Aspartic acid), NGR (Asparagine-Glycine-Arginine) peptide, folic acid (FA) and hyaluronic acid with CPPs is a very effective strategy [84,85,86]. These targeting molecules are usually overexpressed in tumor types, but not in normal cells. Therefore, they can improve the targeting effect for pathological tissues, whilst healthy tissues are not affected by drug delivery. For example, Qi-ying Jiang conjugated the target ligand of FA and the CPP segment of octaarginine (R_8_) to an existing vector (PEI600-CD) composed of β-cyclodextrin and low-molecular-weight polyethylenimine (PEI) to produce a new gene vector FA-PC/R_8_-PC [26]. This vector can form ternary nanocomplexes with pDNA, and further deliver it to tumor sites in vivo with excellent gene transfection efficiency (Figure 3b). Moreover, hyaluronic acid coupled with CPPs can effectively deliver siRNA to macrophages within the atherosclerotic plaques and enhance gene delivery to macrophages in antiatherosclerotic therapy [30], which is a promising nanocarrier for efficient macrophage-targeted gene delivery and antiatherogen (Figure 3c).

In addition to being used as vectors for gene delivery alone, CPPs can also be combined with other non-viral vectors such as liposomes and cationic polymers to achieve high gene transfection efficiency. Integrating different types of functional vectors into one gene delivery system can exert a synergistic effect between the components, improving the low permeability and poor selectivity of CPPs, and so enhance the gene delivery efficiency. Ikramy et al. developed an efficient gene delivery system by combining a CPP segment (R_8_) and pH-sensitive cationic lipid (YSK05) [27]. Positive nanoparticles can be formed by attaching high density R8 to the surface of YSK05 nanoparticles. The particles can further encapsulate pDNA to produce complexes that can lead to high gene transfection efficiency due to the synergistic effect between R_8_ and YSK05. Obdulia and co-workers also developed a gene delivery vector by co-assembly of CPP (WTAS) and a poly β-amino ester (PBAE) polymer [31]. The WTAS-PBAE vector showed high transfection rate, and the results of cell transfection experiments with GL26 cells revealed that WTAS-PBAE vector loaded with GFP pDNA led to virtually complete transfection (> 90%). This excellent transfection efficiency makes it a very promising gene delivery vector for delivering a variety of genetic materials. In addition, the combination of CPPs and inorganic nanoparticles also shows great potential in the application of delivering nucleic acid drugs. For example, Dowaidar et al. found that the conjugation of CPPs-oligonucleotides with magnetic iron oxide nanoparticles can promote cellular uptake of the plasmid and improve the transfection efficiency, which opens up a new way for selective and efficient gene therapy [32].

## 4. Application of Targeted Peptides in Gene Delivery

During gene delivery, an off-target effect may occur when the therapeutic nucleic acids bind to non-specific cells, which is undesirable and will decrease the therapeutic effect of gene therapy. Therefore, selectively delivering vector-nucleic acid complexes to the target cells and exerting the therapeutic effect at specific sites are critical to improve the transfection efficiency of gene therapy [87]. Conjugating targeting ligands such as FA, hyaluronic acid and biomolecules including peptides and proteins can greatly increase the targeting of the gene delivery systems because they can specifically bind to the receptors on cells. Among them, peptides are excellent gene delivery targeting ligands due to their good biocompatibility, ease of synthesis and modification as well as their high response to stimuli. Thus far, more than 700 targeted peptides have been discovered for targeting different cells. The most widely used target peptides among them are NGR and RGD which can specifically recognize tumor angiogenic markers and provide new venues for exploring tumor targeting agents [84].

The NGR motif, whose tumor-targeting ability relies on its specific interaction with CD13 (aminopeptidase N), was identified from a tumor homing peptide. It is often selectively overexpressed in neovascular and some tumor cells, but seldom expressed in quiet vascular endothelial cells. NGR peptides have now been used to promote the targeted delivery of therapeutic agents and enhance antitumor effects [88]. A bi-functional peptide, NGR-10R, which consists of an N-terminal circular NGR motif (CNGRCG) and a C-terminal R_8_ sequence was designed for gene therapy. The R_8_ sequence at the end of NGR-10R can bind to siRNA through electrostatic interaction to form NGR-10R/siRNA nanoparticles. Thanks to the NGR motif, NGR-10R/siRNA nanoparticles can be specifically delivered to MDA-MB-231 cells and localized around the nucleus, thus robustly repressing gene expression in MDA-MB-231 and HUVEC (a CD13^+^/α_v_β_3_^+^ cell) (Figure 4a) [28]. In the study of Yang, as a targeted peptide, NGR plays a navigational effect, enabling the pcCPP/NGR-LP dual-modified liposomes vector to accumulate at the tumor site. Finally, with the aid of CPPs, the siRNA-loaded vector enters target cells efficiently [33]. In addition to targeting siRNA to MDA-MB-231 cells, the NGR motif can effectively deliver siRNA to HT-1080 cells and downregulate target genes with the synergistic effect of other vectors. Chen et al. designed the LPD-poly(ethylene glycol) (PEG)-NGR vector by modifying PEGylated LPD using the NGR motif. It can target CD13 expressed in the tumor cells or tumor vascular endothelium, effectively delivering siRNA to the cytoplasm of HT-1080 cells and silence the target gene [34].

Different from NRG, the RGD peptide can specifically bind to integrin in tumor endothelial cells and act as ligand to target tumor cells that overexpress α_v_β_3_ integrin [89,90]. As an attractive tumor cell receptor, integrin plays a major role in promoting the proliferation, migration, invasion and survival of tumor cells. Therefore, gene vectors modified by RGD peptide can block cell–cell and cell-matrix adhesions by competing with adhesion proteins for cell surface integrins, thus achieving targeted selectivity to tumor cells and improving the efficiency of gene transfection. In view of this, a large number of RGD peptide-based gene vectors have been developed. Recently, lung cancer and bronchial cancer have become the most deadly cancers due to the aggravation of air pollution. In order to develop new targeted, effective and less painful therapies, Yang et al. synthesized the RRPH (RGD-R_8_-PEG-HA) which is composed of peptide (RGD-R_8_) and PEGylation on HA to coat PFC (plasmid complex). The obtained RRPHC nanoparticles (RRPH coated PFC complex) achieve long-term circulation and tumor tissue-penetration while maintaining the high transfection efficiency of PFC [29]. Kim et al. designed a targeted gene vector, RGD/PEI/WSC, which can combine the RGD to chitosan and PEI, for α_V_β_3_ integrin-overexpressing tumor cells [35]. In vivo experiments show that the vector can suppress the growth of PC3 prostate tumor cell xenograft model by silencing BCL2 mRNA, which is expected to be a good candidate for a specific targeted gene vector without cytotoxicity (Figure 4b).

Oncolytic adenovirus has been widely used in clinical trials of cancer gene therapy [91,92]. Moreover, tumor targeted gene virus therapy (CTGVT) may be an effective strategy for the treatment of advanced or metastatic cancer [93]. In a previous study, Luo et al. found that replicating adenovirus (AD-ZD55-miR-143) showed specific anti-rectal cancer efficacy in vitro. However, its anti-tumor effect in vivo is not ideal, because the vector does not increase the chance of reaching target cells. To solve this problem, they developed AD-RGD-survivin-ZD55-miR-143, a novel triple regulatory oncolytic adenovirus which significantly enhanced the anti-tumor effect and directly broadened the treatment options for colorectal cancer [36]. RGD peptides with a circular structure, i.e., cyclic (c)RGDs can also be used for tumor targeting studies—being more active due to their conformation-less assembly than linear RGD oligopeptides. Moreover, (c)RCDs are resistant to proteolysis and have higher affinity to integrin receptors [94]. Therefore, many five membered ring RGDs containing pentapeptides have been used to endow gene vectors with tumor targeting [95]. Alam et al. reported that cRGDs can selectively enter cancer cells overexpressing α_v_β_3_ integrin carrying siRNA for gene silencing [38]. A further study indicated that cRGDs can specifically guide siRNA to cells expressing α_v_β_3_, resulting in effective knocking out of selected genes and significantly reducing tumor growth [39]. In addition, cRGDs were employed to promote cellular internalization of polyplex micelles encapsulating anti-angiogenic pDNA by tumor vascular endothelial cells, which abundantly express RGD-specific α_v_β_3_ and α_v_β_5_ integrin receptors and thereby exhibit anti-tumor activity against pancreatic adenocarcinoma upon systemic injection [96,97]. Moreover, liposomes modified with cRGD peptide can be used to deliver drugs to targeted cancer cells [40].

Our group is also devoted to designing peptide carriers with targeting functions. Recently, we have designed an amphiphilic peptide Ac-RGDGPLGLAGI_3_GR_8_-NH_2_ with two charged chain segments distributed at the end and a hydrophobic chain segment in the middle [37]. It can selectively kill cancer cells through the specific recognition and binding of RGD fragments to cancer cell membranes and cleavage of PLGLA fragments by tumor-overexpressed matrix metalloproteinase-7 enzymes. The R_8_ sequence can induce efficient condensation of DNA into dense nanoparticles, resist enzymatic degradation of DNA, ensure successful delivery of DNA into cells, and improve the expression level as well as transfection rate of target genes [87]. Moreover, we also combined the cRGD peptide to gold nanoparticles (AuNPs) which has been widely used in the delivery of nucleic acid molecules due to its good biocompatibility and easy surface functionalization [98,99]. We designed the peptide of sequence (CRGDKGPDC)GPLGLAGIIIGRRRRRRR-NH_2_ (CPIR28) which was grafted onto the surface of AuNPs by the one-pot synthesis method [41]. The CPIR28-AuNPs nanocomposite can effectively condense DNA and improve the intracellular transport of genes (Figure 4c).

## 5. Application of Membrane Active Peptides in Gene Delivery

After cell uptake, successful release of vector/nucleic acid complexes from endosomes is a major obstacle for effective gene therapy. After the vector/nucleic acid complexes cross the membrane barrier into the cell through endocytosis, vesicles will enclose them and develop into early endosomes, which then mature to form late endosomes and then fuse with lysosomes. In order to exert the therapeutic effect of nucleic acid drugs, the complexes need to escape from the endosomes and enter into the cytoplasm. Otherwise, the nucleic acid drugs will be degraded by hydrolases [46]. Therefore, developing vectors with endosomal escape ability is essential for efficient gene delivery. There are two ways to achieve endosomal escape. First, considering the acidic environment inside the endosomes, materials with a buffer effect in the acidic environment, such as chlorine and calcium, can be added to assist endosomal escape. These buffer agents can prevent endosomes from binding to lysosomes, vacuolate endosomes and then decrease the membrane stability. However, these chemicals are generally only used in vitro and not suitable for clinical applications due to their potential cytotoxicity. Nevertheless, the acidic endosomal environment suggests that we can introduce amino acids with a acidic buffering effect into the carrier to destroy the endosome membrane by proton pump for the purpose of endosomal escape. Since only histidine has a buffering effect among the 20 common amino acids due to its imidazole group, it is often embedded into the carrier to improve endosomal escape during delivery of nucleic acids. RALA, which is a 30-mer cationic amphipathic helical peptide, contains seven hydrophilic arginine residues on one side of the helix, and hydrophobic leucine residues on the other side. When the pH drops, the α-helicity of RALA increases to achieve endosomal escape and release of the cargo [42]. Therefore, Vimal K et al. used RALA peptides to condense mRNA and effectively deliver them to dendritic cells [43]. Subsequently, the RALA-mRNA nanocomplexes successfully escaped from endosomes and expressed mRNA in the cell cytosol to promote antigen specific T cell proliferation as well as evoking T cell immunity in vivo (Figure 5a). In addition to delivering mRNA, RALA can also deliver siRNA with high efficiency. Eoghan J. Mulholland et al. reported that RALA is an effective siRNA carrier targeting the FK506-binding protein and has great potential in promoting angiogenesis for advanced wound healing applications (Figure 5b) [44]. Recent studies have found that the introduction of histidine into RALA peptide can further improve the endosomal escape ability of the vectors, thereby increasing the transfection efficiency. For example, Liu et al. designed a new peptide-based vector HALA2 with ability of endosomal escape and high cell transfection efficiency by adjusting the ratio of histidine and arginine in the RALA peptide [45]. HALA2 replaced two arginines close to the C-terminal of RALA with histidine, which reduced the number of positively charged amino acids in HALA2 from 7 to 5, resulting in a better transfection rate than RALA. In addition, introducing histidine fragments into other kinds of vectors can also improve their endosomal escape ability. Chitosan has the advantages of non-toxicity, non-immunogenicity, biodegradability and good biocompatibility as a gene vector. However, chitosan cannot mediate the escape of endosome due to its low endosomal escape rate and poor buffer capacity. For this reason, Liu et al. introduced histidine into chitosan and obtained a new vector with good solubility, strong binding ability to siRNA and excellent endosomal escape performance [100].

Secondly, using membrane active peptides with membrane destruction capability to destroy the endosomal membrane can also realize endosomal escape and release the vector/nucleic acid complex into the cytoplasm. Recently, a series of membrane active peptides have been designed. For example, (LLHH)_3_ and (LLKK)_3_-H_6_ are two typical amphiphilic membrane active peptides that can destroy endosomal membranes and regulate the “proton sponge effect”. Introducing them into vectors containing rigid acyl and polyarginine, Yang et al. designed two multifunctional peptide vectors, C_18_-C(LLKK)_3_-H_6_-R_8_ and C_18_-C(LLHH)_3_C-R_8_. They found that each functional fragment showed a synergistic effect, and the presence of membrane active peptide significantly improved the endosomal escape efficiency and transfection rate, which greatly promotes the application of peptide-based vectors in the treatment of genetic diseases [46]. In the past few years, Bechinger and co-workers have been devoted to developing pH-responsive cationic amphiphilic membrane active peptides rich in histidine residues for gene delivery. They have designed a variety of LAH4-based peptides which have been proven to be able to bind to plasmid DNA and facilitate its cellular uptake and endosomal escape [47,101,102,103]. Among them, some derivative peptides of LAH4 not only have the ability to bind to plasmid DNA, but also have strong siRNA and mRNA delivery capabilities [47]. To date, the interactions of LAH4-based peptides and bio-membrane have been studied in detail by biophysical methods, and the results indicate that these peptides show strong delivery capacity for a variety of cargoes, including nucleic acids, peptides and proteins [104]. The histidine-rich amphiphilic peptide KH27K has also been developed as a “proton sponge” escape endosomal agent. Unlike LAH4, KH27K is currently mainly used to deliver virus particles into the cell to achieve the intracellular release of the virus, and this “membrane release” activity is consistent with its pH dependent hemolysis activity. However, there is no clear study on the intracellular delivery of nucleic acid molecules [48,49].

Antibacterial peptides (AMPs) with an α-helical amphiphilic structure can also effectively promote endosomal escape. They are primarily found in bacteria and have activities against a variety of microorganisms. Most of them are composed of nearly 50% hydrophobic residues and are usually positively charged due to the presence of lysine and arginine fragments. The spatially separated hydrophobic and charged regions endow them with membrane interaction activity. In view of the characteristics of AMPs, Cirillo et al. designed a short cationic amphiphilic α-helical peptide G(IIKK)_3_I-NH_2_ with endosomal escape ability and high affinity towards colon cancer cells [50]. They report that when interacting with negatively charged DPPG small unilamellar vesicles, the peptides fold into α-helical structure helping to carry nucleic acids across the cell membrane and achieving endosomal escape, thus enabling the protection and selective delivery of siRNA to cancer cells (Figure 5c). Melittin is a multifunctional AMP that can inhibit many Gram-negative and Gram-positive bacteria. It is widely used to facilitate the endosomal escape of nanoparticles because of its significant cleavage activity in mammals both in vivo and in vitro. However, this amphiphilic peptide from bee venom has obvious toxicity to mammalian cells. If it is directly used to deliver nucleic acids, the transfection efficiency will be reduced due to the increase of cytotoxicity [105]. Therefore, melittin analogues have been designed in order to reduce the toxicity while promoting the ability to promote endosomal escape [106]. Glutamic acid and histidine residues on peptides are negatively charged due to deprotonation in the extracellular medium; however, in endosomes with a pH of about 5, the two amino acids are protonated, which reduces the hydrophilicity of the peptide and exposes its cleavage activity. Therefore, the method of replacing the basic amino acids in melittin with glutamic acid or histidine can be used to enhance the cleavage ability of the pH sensitive peptide. In views of this principle, a series of novel pH-sensitive peptides have been developed. Melittin analogues such as CMA-1, CMA-2, CMA-3, CMA-4, NMA-3 [52] and acid-melittin [51] have been obtained and used to conjugate with PEI to improve the intracellular endosomal escape of the PEI/DNA complex. Compared with CMA-1-PEI and CMA-4-PEI that covalently linked PEI to the N-terminal of peptide, C-terminal modified CMA-2-PEI, CMA-3-PEI and acid-melittin-PEI complexes showed strong cleavage activity at pH 5. The transfection experiments also showed that CMA-2-PEI and CMA3-PEI complexes induced significant gene expression [53,54]. Not all N-terminal modified melittin analogues have poor cleavage ability. For example, Kloeckneret al. proved that the transfection efficiency can be significantly improved by introducing N-terminal PEI-coupled melittin analogue NMA-3 into the EGF/OEI-HD-1 complex gene vector [52]. In addition, considering the effect of glutamate replacement location on peptide cleavage activity, Tamemoto et al. designed four melittin analogues and studied the optimal position of glutamate substitution. The results showed that a novel attenuated cationic cleavage peptide MEL-L6A10 with higher delivery activity, relatively lower cytotoxicity and higher endolytic activity can be designed by placing Glu on the boundary of the hydrophobic/hydrophilic region [55]. RV-23 is a pH-sensitive endolytic peptide extracted from Rana Linnaeus. Zhang et al. obtained a pH-sensitive endolytic peptide by replacing the positive charge residues in RV with glutamate. This substituted RV-23 peptide can promote the obvious destruction of cell intima and promote the entry of the carrier/nucleic acid complex into the cytoplasm. Thus, the gene transfection rate was significantly increased and the PEI-mediated cell transfection rate promoted [53].

## 6. Application of NLS Peptides in Gene Delivery

In gene delivery, some nucleic acid drugs, such as siRNA and mRNA, can directly play a therapeutic role in the cytoplasm after endosomal escape. However, for pDNA, DNA needs to be further transferred into the nucleus to realize its therapeutic effect. In such cases, whether DNA can be assisted to enter the nucleus is a key factor to evaluate the delivery capacity of non-viral gene vectors [56]. Macromolecules such as proteins cannot directly enter the nucleus due to the strong impedance from the nuclear envelope, and their transport into the nucleus must be regulated by the nuclear pore complex (NPC) [107,108]. When the protein enters the nucleus, the NLS (a short cationic peptide sequence) on the proteins can be recognized by the corresponding nuclear transporter, which helps them reach the nucleus through NPC with the assistance of transporter and nucleoporin [109,110]. Based on this, introducing NLS peptide sequences into non-viral vectors may achieve efficient delivery of the therapeutic DNA into the nucleus. Generally, NLS peptides can be divided into two categories, termed monopartite NLS (MP NLS) and bipartite NLS (BP NLS). The MP NLS is a single cluster composed of 4–8 basic amino acids, and the most common MP NLS is the basic heptad-peptide derived from SV40 virus large T antigen. Since this NLS is only related to nuclear transport and has no effect on improving cell uptake, it needs to enter the cytoplasm first to assist gene drugs to enter the nucleus [57]. The MP NLS peptides are often combined with CPPs to fabricate vectors which can promote transmembrane transport, nuclear localization and further realize targeting delivery of pDNA. For example, Yan et al. constructed a new nucleus-targeted NLS (KALA-SA) vector by combining MP NLS, KALA (a cationic CPP) and stearic acid (SA). Besides enhancing cytoplasmic transport, this vector realized targeting localization and provided a promising strategy for the treatment of lung cancer [56]. Moreover, conjugating MP NLS peptide with targeted peptide RGD can also achieve an excellent therapeutic effect. Following this strategy, Ozcelik modified MP NLS peptide and RGD peptide onto AuNPs with radio-sensitizer ability to initiate X-ray radiation-induced cell death and achieve the effect of killing or inhibiting cancer cells while retaining the normal cells. Interestingly, the results indicated that AuNPs with both cancer cell targeting and nuclear targeting capabilities are far more specific and lethal than AuNPs modified by NLS or RGD alone [58]. In order to significantly improve the delivery capacity, Hao et al. integrated NLS with CPPs (TAT) and RGD (REDV) with a selectively targeting function for endothelial cells to obtain the REDV-TAT-NLS triple tandem peptides [59]. By inserting glycine sequences with different repeats into the triple tandem peptides, the functions of each peptide were synergistically performed. The peptide complexes can be used as vector to deliver pZNF580 plasmid in endothelial cells, which can significantly improve the revascularization ability of human umbilical vein endothelial cells in vitro and in vivo, thus providing a promising and effective delivery option for angiogenesis treatment of vascular diseases (Figure 6a). Recent studies revealed that Mice Fibroblast Growth Factor 3 (FGF3) is a peptide containing multiple NLS peptides. RLRR and RRRK are two peptide sequences that can induce nuclear localization in this NLS. Introducing the RRRK peptide fragment into PAMAM non-viral vectors can significantly improve the transfection efficiency and gene expression of the vectors [61]. In addition, using four NLS derived from SV40 virus with glycine residues as spacers, Ritter synthesized the NLS tetramer of SV40 large T antigen. This lysine-rich peptide solves the past problem of NLS interfering with gene expression by covalent binding to nucleic acid molecules: it binds and concentrates nucleic acid molecules by electrostatic interaction to form stable polymers with nuclear transport properties [62]. More importantly, NLS has also been used in clustered regularly interspersed short palindromic repeats (CRISPR)/CRISPR-associated protein 9 (Cas9) gene editing technology which is widely studied nowadays. As a nuclear targeting peptide, NLS can specifically transport the vector into the nucleus, so that the Cas9/sgRNA plasmids can be accurately delivered to the tumor sites. Studies have shown that combination of NLS peptides with other non-viral vectors can significantly improve the gene editing ability of Cas9/sgRNA. For example, using NLS peptide and AS1411 aptamer as delivery vector, Cas9/sgRNA can achieve effective genome editing in targeted tumor cells [60], down-regulate the expression of FAK protein in tumor cells, and thus lead to tumor cell apoptosis (Figure 6b).

In addition to adding MP NLS to various nonviral vectors to achieve efficient nuclear delivery of therapeutic DNA, BP NLS composed of two or more positively charged amino acid clusters have also been developed and used for gene delivery. Matschke synthesized a modified NLS dimer structure, NLS-Ku7O_2_. Highly efficient nuclear transport and transgenic expression were realized by co-assembling this BP NLS-Ku7O_2_ with PEI and DNA into a ternary gene carrier complex [63].

## 7. Application of Other Peptides in Gene Delivery

To date, great success has been achieved in developing nonviral vectors using materials including peptides, proteins, dendrimer and liposomes. Although the gene transduction efficiency has been improved, the gene expression level is still far lower than that of viral vectors and cannot meet the clinical requirements. However, the inherent toxicity, immunogenicity and complex preparation process of viral vectors greatly limit their clinical application [64]. Therefore, great efforts have been devoted to building supramolecular assemblies that can simulate both the viral structure and function. The therapeutic nucleic acids are encapsulated into these supramolecular assemblies and delivered into cells, in the hope of obtaining efficient gene delivery vectors while reducing the inherent risk of viruses [111,112,113]. Recently, because of the good biocompatibility and low cytotoxicity of peptides, more and more research has been focused on imitating the virus structure through the co-assembly of peptide and nucleic acid [114,115]. Spherical viral capsids have discrete nanospace, good cell transfection ability and biodegradability, and can therefore be used as nanocarriers for nucleic acid drug delivery [116,117,118]. Inspired by the spherical virus, Matsuura found that the 24-mer β-annulus peptide involved in dodecahedral skeleton formation of tomato bushy stunt virus can spontaneously assemble into a “spherical artificial virus-like capsid” with a size of 30–50 nm. The cationic interior of the artificial viral capsid is hollow, allowing DNA molecules to be effectively encapsulated [65,66]. Based on the above, Matsuura K. used β-cyclic GGGCG peptide as the binding site of AuNPs, which finally self-assembled into nanocapsules with a diameter of 50 nm. This strategy extends the design of artificial viral capsids and can be further used for the delivery of nucleic acid molecules [67]. The short peptide H4K5HC_BZl_C_BZl_H obtained by rational design is also a spherical viral capsid. Compared with the past research on spherical artificial viruses, this spherical viral capsid has a low aspect ratio because of adding the cysteine in the center of the short peptide H4K5HC_BZl_C_BZl_H. This nanostructure can not only mimic the sequential decomposition of spherical viruses in response to stimuli, but also simulate the complex morphology and intracellular transformation of spherical viruses, making it an effective DNA delivery vector [68]. In addition to spherical artificial virus particles, filamentous, rod-shaped and cocoon-like virus particles have also been developed as artificial viruses. For example, the short peptide K3C6SPD which contains three fragments including N-terminal cationic fragment, β-sheet forming fragment and C-terminal hydrophilic fragment can be co-assembled to obtain cocoon-like artificial virus particles (Figure 7a) [69,70]. Ruff designed triblock molecules SP-CC-PEG which can self-assemble into mushroom nanostructures [71]. Using self-assembled non-centrosymmetric nanostructures similar to supramolecular mushrooms as caps, virus-like particles with a certain length are created and then wrapped on DNA to generate filamentous particles (Figure 7b). Marchetti designed a triblock peptide C−S10−B containing a segment of artificial lysine capsid using a de novo design method. Through electrostatic interaction, it interacted with the phosphate of single stranded or double stranded DNA and co-assembled into coronavirus-like particles, mimicking the corresponding function of viral capsid proteins [119]. These theoretical studies provide new ideas for current nucleic acid delivery.

The efficient delivery of nucleic acids has been achieved by constructing new nonviral delivery systems using single or several lysines as functional fragments. Furthermore, many studies have shown that cationic poly(l-lysine) (PLL) can also be used to achieve efficient nucleic acid transport in vivo. PLL can mediate condensation of anionic nucleic acids to form smaller nanoparticles and protect them from enzymatic and physical degradation [120]. Yugyeong Kim et al. synthesized a new cationic AB2 miktoarm block copolymer consisting of two cationic PLL blocks and one PEG block, which can form effective nanocomplexes with pDNA. The nanocomplexes can release pDNA effectively under reducing conditions and show high level of gene expression [121]. However, for PLL, its in vitro transfection efficiency is poor in the absence of any covalently attached functional moieties to promote gene targeting or uptake [120]. To solve this problem, researchers have discovered a new cationic poly-amino acid, that is, poly(l-ornithine) (PLO). Compared to PLL that contains a tetramethylene spacer, PLO possesses a trimethylene unit in the side chain. It can complex with pDNA or mRNA and enhance transfection efficiency [122]. One big issue of nonviral gene delivery is unnecessary uptake by the reticuloendothelial system, mainly the liver. In general, 60–70% of nucleic acid molecules are taken up by scavenger receptors on liver Kupffer cells when being injected into the body without the protection of carrier molecules. This nonspecific scavenging behavior results in a significant reduction in the efficiency of drug entry into target tissues [123]. Lysine polymer exhibits excellent potential in solving this problem by avoiding unwanted uptake by the reticuloendothelial system. Recently, Anjaneyulu Dirisala et al. found that oligo(l-lysine) conjugated linear or two-armed PEG can transiently and selectively mask liver scavenger cells, effectively inhibiting sinusoidal clearance of nonviral gene carriers, thereby increasing their gene transduction efficiency in target tissues [124].

The formation of artificial viruses is based on the non-covalent interaction of peptide/peptide or peptide/DNA. By rational design of the peptide structure, the morphology, stability and transfection efficiency of the peptide/DNA hybrid structure can be regulated to construct artificial viruses [125]. In recent years, our group has been focusing on the design and study of different surfactant-like peptides to induce effective DNA condensation and so produce artificial viruses for protecting DNA from enzymatic degradation. For example, we designed six surfactant-like peptides with the same amino acid composition but different primary sequences. Because the peptide residues have different side chain size and hydrophobicity, this can lead to different self-assembled structures [126]. Among them, I_3_V_3_A_3_G_3_K_3_ is a dumbbell-like peptide which can effectively induce DNA condensation into a virus-like structure through non-covalent interactions such as electrostatic interaction, hydrophobic interaction and hydrogen bonding [72]. The final formed structure can imitate the essence of a viral capsid to condense and wrap DNA, which is conducive to effective gene delivery in the later stage (Figure 7c). AKAEAKAE, another peptide segment we designed, has strong β-sheet forming capability and can co-assemble with PNA to obtain peptide nucleic acid-peptide conjugate, T′_3_(AKAE)_2_. It can condense DNA at low micromole concentrations, which suggests it can be a gene delivery vector [112,127]. NapFFGPLGLAG(CK_m_)_n_C peptides have been developed by introducing several functional segments, that is, an aromatic segment of Nap-FF to promote peptide assembly by providing hydrophobic interaction, an enzyme-cleavable segment of GPLGLA to target cancer cells, and several positively charged K residues for DNA binding. These peptides can self-assemble into homogenous capsid-like nanospheres with high stability under the synergy of functional segments [64]. Moreover, they can further co-assemble with DNA to protect the genome from enzymatic digestion and greatly improve the efficiency of gene delivery (Figure 7d).

## 8. Concluding Remarks and Future Perspectives

Developing versatile vectors to deliver therapeutic nucleic acids into target cells/tissues is critical for gene therapy. As promising candidates, peptide-based vectors have been widely used for delivering therapeutic nucleic acids. In addition to condensing nucleic acids to form nanoparticles for protecting them from being degraded by enzymes, the rationally designed functional peptides can also help to overcome a series of biological barriers including crossing cell membrane, escaping from endosome, entering the nucleus, etc., and finally release the therapeutic nucleic acids at the target sites. These functional peptides can not only be used alone to overcome such biological barriers in gene delivery, but also can be combined to form multifunctional peptide vectors. Moreover, they can also be introduced into other nonviral gene delivery systems as functional elements to enhance the delivery capacity, which greatly expands the application of peptides in gene therapy. However, it is worth noting that although there have been a large number of reports on peptide-based gene delivery systems, most of them are still in the stage of theoretical research and animal experiments, and there are still many challenges before peptide vectors being considered for clinical use. First, the peptide-based vectors often suffer from short circulating half-time and poor chemical/physical stability, which greatly hinder the use of peptide–nucleic acid complexes in clinical trials. Effective strategies such as modifying the peptides with unnatural amino acids should be developed to improve the structural stability of the peptide-based gene delivery systems. Secondly, although peptide sequences with different functions can be combined to overcome various barriers for efficient gene delivery, this approach carries the risk of reducing individual functions. Therefore, the combination of peptide with other components without affecting the function of each part is still a problem to be solved. Thirdly, how to precisely control the microstructures of the peptide–nucleic acids complexes so as to achieve effective cellular uptake and gene transfection at targeted sites is another important issue. Modifying the peptidic vectors with stimulus-responsive fragments to design smart delivery systems so that they can perceive changes in the disease microenvironment and trigger gene release may be an effective way to solve this problem. In summary, although there has been much study and great success in the field of peptide-based gene vectors, researchers still need to move forward to find solutions for promoting peptidic gene delivery systems for them to become a gene therapy product that can be approved for clinical applications. Such research would not only promote the rapid development of peptide-based gene delivery systems, but also enable some emerging gene therapy strategies, such as CRISPR/CAS9 technology and mRNA vaccines to be applied in the human body at an early date.

## Figures and Tables

**Figure 1 nanomaterials-12-04076-f001:**
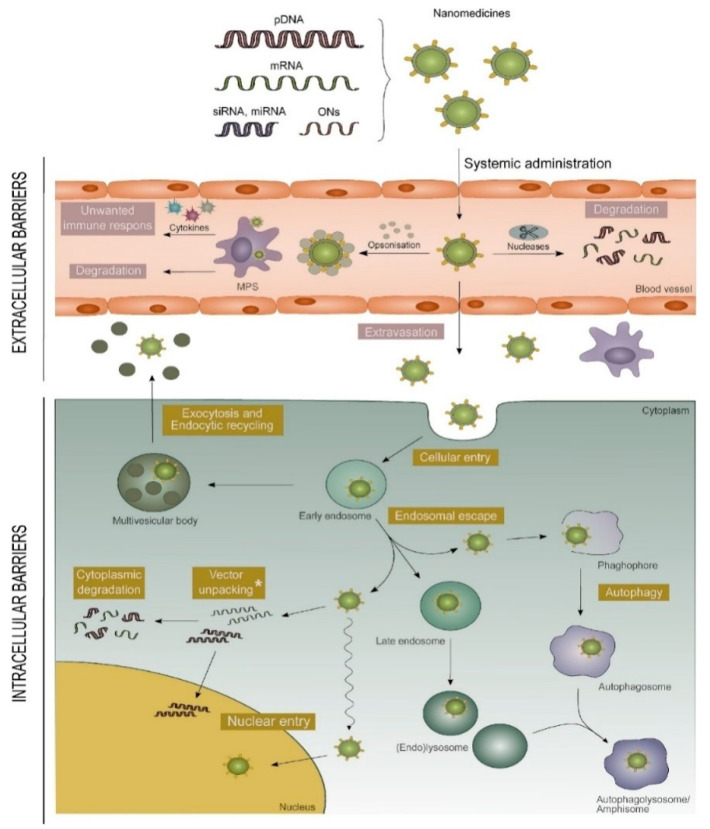
Biological barriers to overcome when using nonviral vectors to deliver nucleic acids in vivo. Nucleic acids bind to peptides through electrostatic interactions, transferring them across the cell membrane via endocytosis, providing endosome escape, and ultimately releasing the associated nucleic acids in the cytoplasm or nucleus. Reprinted with permission from Ref. [20]. Copyright 2018, Elsevier.

**Figure 2 nanomaterials-12-04076-f002:**
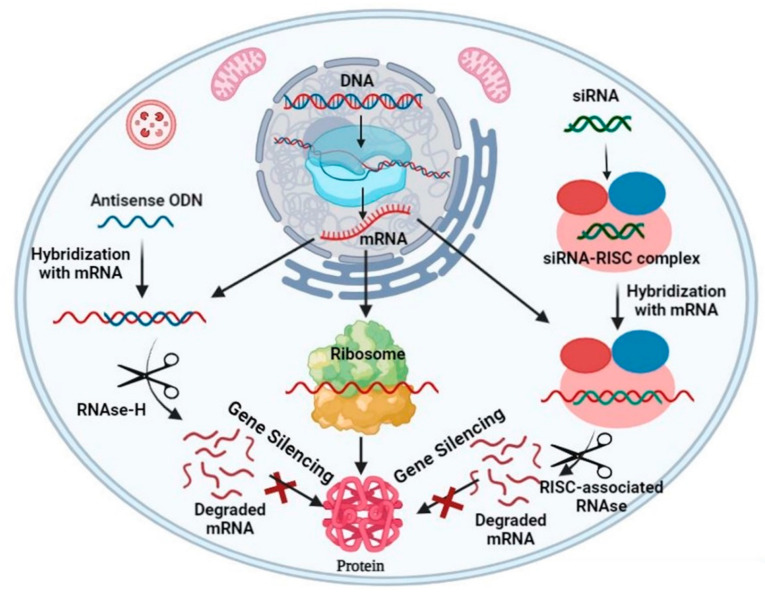
Schematic illustration of different cellular pathways involved in gene silencing. Reprinted with permission from Ref. [21]. Copyright 2022, Elsevier.

**Figure 3 nanomaterials-12-04076-f003:**
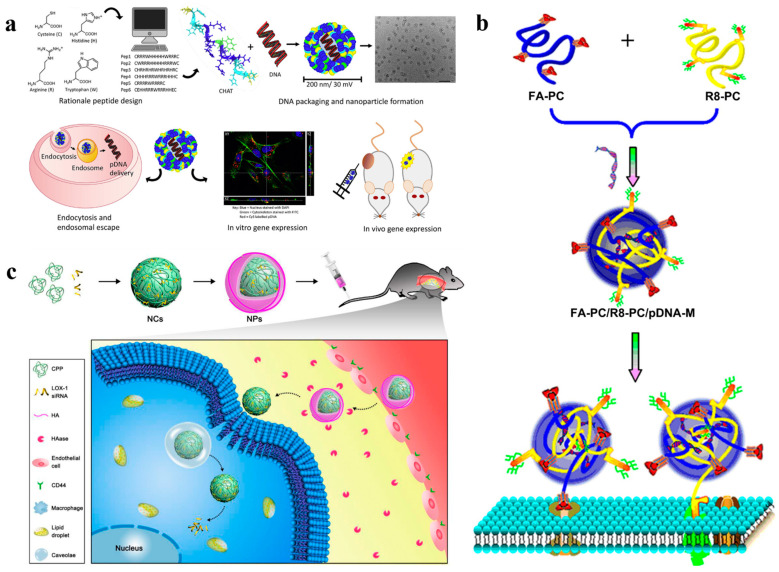
(**a**) CHAT peptide condenses pDNA to produce cationic nanoparticles less than 200 nm in diameter. The complex can cross the cell membrane through endocytosis and successfully escape from the endosomes, obtaining high transfection efficiency. Reprinted with permission from Ref. [22]. Copyright 2020, Elsevier. (**b**) The process of preparation of nanoparticles formed from FA-PC/R8-PC/pDNA complex. Reprinted with permission from Ref. [26]. Copyright 2011, Elsevier. (**c**) CPPs condense siRNA and deliver it to macrophages. Reprinted with permission from Ref. [30]. Copyright 2018, American Chemical Society.

**Figure 4 nanomaterials-12-04076-f004:**
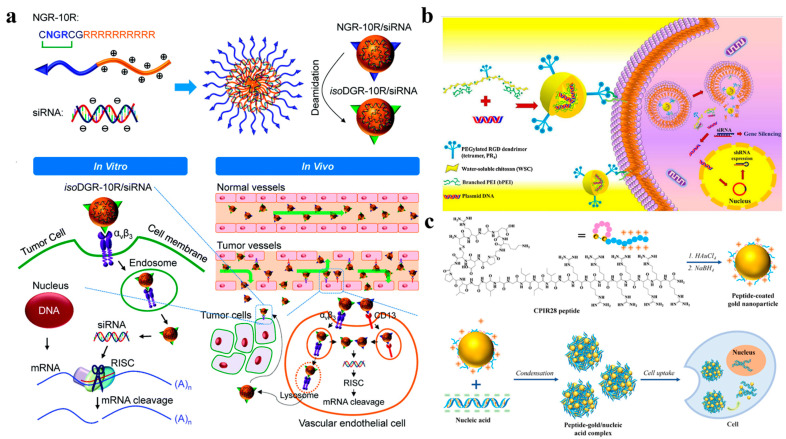
(**a**) Bi-functional NGR-10R peptide condenses siRNA to form spherical nanostructures which can enter cells by receptor α_v_β_3_ and CD13 mediated endocytosis. After escaping from the endosomes/lysosomes, siRNA is released into the cytoplasm and loaded by the RISC. Reprinted with permission from Ref. [28]. Copyright 2015, Biomaterials Science. (**b**) RPgWSC-pDNA complexes can suppress solid tumor growth by silencing BCL2 mRNA. Reprinted with permission from Ref. [35]. Copyright 2017, Elsevier. (**c**) CRIP28-AuNPs form nanocomplexes with nucleic acids by electrostatic interaction for cellular delivery. Reprinted with permission from Ref. [41]. Copyright 2022, Elsevier.

**Figure 5 nanomaterials-12-04076-f005:**
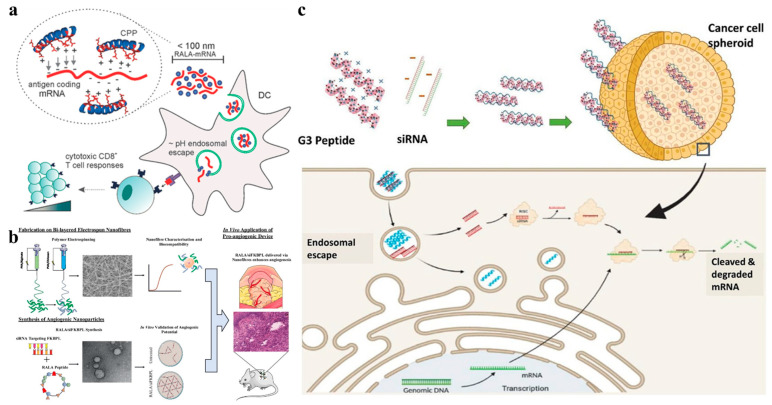
(**a**) RALA peptides condense mRNA into nanoparticles, releasing mRNA in dendritic cell cytosol to promote antigen specific T cell proliferation. Reprinted with permission from Ref. [43]. Copyright 2017, Wiley Online Library. (**b**) RALA peptides form a complex with siRNA to deliver siRNA into the cell and promote the regeneration of blood vessels. Reprinted with permission from Ref. [44]. Copyright 2019, Elsevier. (**c**) G3 peptide was assembled with siRNA and delivered to cancer cells, where siRNA was released to regulate gene expression in cancer cells. Reprinted with permission from Ref. [50]. Copyright 2021, American Chemical Society.

**Figure 6 nanomaterials-12-04076-f006:**
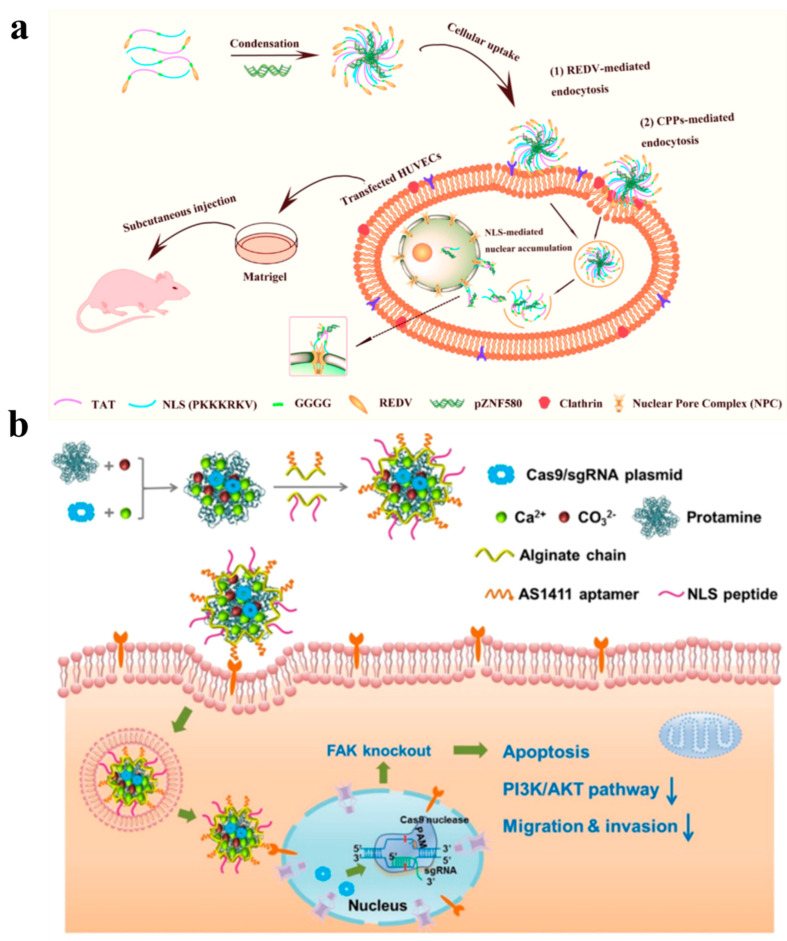
(**a**) The REDV-G4 -TAT-G4-NLS peptide assembles with pZNF580 plasmid to form nanocomplexes, which are transported to endothelial cells by the targeting effect of REDV. After the transmembrane and endosomal escape, the complexes enter the nucleus by the action of NLS to promote the expression of pZNF580 plasmid and enhance the revascularization ability of cells. Reprinted with permission from Ref. [59]. Copyright 2017, American Chemical Society. (**b**) The Cas9/sgRNA plasmid gene delivery system was prepared by the self-assembly method, which can specifically deliver the plasmid to the nuclei of tumor cells by the targeting of NLS, and knock down the protein tyrosine kinase 2 (PTK2) gene to the down-regulated local adhesion kinase (FAK). Reprinted with permission from Ref. [60]. Copyright 2019, American Chemical Society.

**Figure 7 nanomaterials-12-04076-f007:**
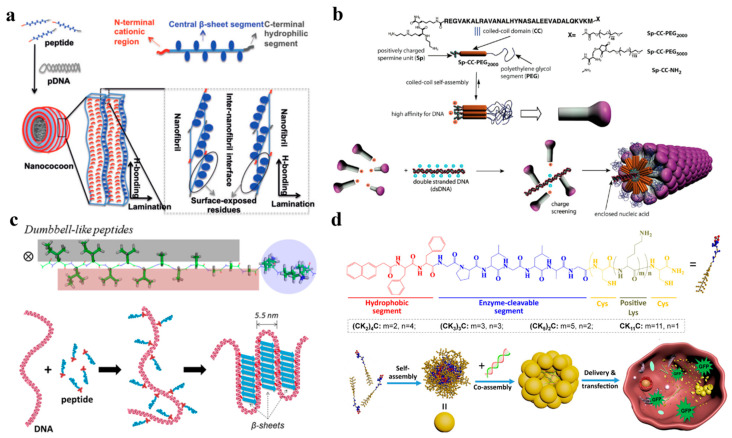
(**a**) Co-assembly of the K3C6SPD short peptide with plasmid DNA develops cocoon-like viral mimics. Reprinted with permission from Ref. [70]. Copyright 2017, German Chemical Society. (**b**) The mushroom shaped nanostructures SP-CC-PEG created by the synergistic self-assembly of three functional fragments, which has high affinity with DNA by electrostatic interaction, is used to prepare synthetic filamentous viruses. Reprinted with permission from Ref. [71]. Copyright 2013, American Chemical Society. (**c**) The dumbbell-like peptide, I_3_V_3_A_3_G_3_K_3_, binds onto the DNA chain through electrostatic interactions, and then self-associates into β-sheets under hydrophobic interactions and hydrogen bonding, the resulting final formed structure being able to imitate the essence of viral capsid to condense and wrap DNA. Reprinted with permission from Ref. [72]. Copyright 2018, American Chemical Society. (**d**) NapFFGPLGLAG(CK_m_)_n_C peptides, containing the multifunctional segment, self-assemble into stable nanospheres which can encapsulate DNA by interacting with DNA in the interior, and finally realize intracellular delivery and release of genome. Reprinted with permission from Ref. [64]. Copyright 2022, Elsevier.

**Table 1 nanomaterials-12-04076-t001:** Types of peptides designed for use in gene delivery.

Peptide Type	Name	Sequence *^a^*	Reference
CPPs	CHAT	CHHHRRRWRRRHHHC	[22]
	LH2	Ac, T, C-LHHLCHLLHHLCHLAG Ac-GALHCLHHLLHCLHHL Ac -LHHLCHLLHHLCHLGA Ac -LHHLCHLLHHLCHLGA	[23,24]
	SRCRP2-11 SRCRP2-11-R	GRVEVLYRGSW GRVRVLYRGSW	[25]
	R_8_	RRRRRRRR	[26,27,28,29]
	Penetratin	RQIKIWFQNRRMKWKK	[30]
	WTAS	PLKTPGKKKKGKPGKRKEQEKKKRRTR	[31]
	PF14	Stearyl-AGYLLGKLLOOLAAAALOOLL-NH2	[32]
	CPP	CGRRMKWKK	[33]
Targeted peptides	circular NGR	CNGRCG	[28]
	NGR	NGR	[33,34]
	RGD	RGD	[29,35,36,37]
	Trivalent cRGD	HCACAE[cyclo(RGD-d-FK)]E[cyclo(RGD-d-FK)]_2_	[38]
	cRGD	cyclo(RGD-d-FK)	[39,40]
	cyclic iRGD	cyclo (CRGDKGPDC)	[41]
Membrane active peptides	RALA	WEARLARALARALARHLARALAHALHACEA	[42,43,44]
	HALA2	WEARLARALARALARHLARALAHALHACEA	[45]
	(LLHH)3	CLLHHLLHHLLHH	[46]
	(LLKK)3-H6	LLKKLLKKLLKKCHHHHHH	[46]
	LAH4	KKALLALALHHLAHLALHLALALKKA	[47]
	KH27K	KHHHHHHHHHHHHHHHHHHHHHHHHHHHK	[48,49]
	G3	GIIKKIIKKIIKKI	[50]
	Melittin	GIGAVLEVLTTGLPALISWIEEEEQQ	[51]
	CMA-1	EEGIGAVLKVLTTGLPALISWIKRKRQQC	[52]
	CMA-2	GIGAVLKVLTTGLPALISWIHHHHEEC	[53,54]
	CMA-3	GIGAVLKVLTTG LPALISWIKRKREEC	[54]
	CMA-4	EEGIGAVLKVLTTG LPALISWIHHHHQQC	[52]
	NMA-3	CGIGAVLKVLTTGLPALISWI KRKREE	[52,53]
	acid-Melittin	GIGAVLKVLTTGLPALISWIKRKRQQ	[51]
	Mel-L6A10	GIGAIEKVLETGLPTLISWIKNKRKQ	[55]
	RV-23	RIGVLLARLPKLFSLFKLMGKKV	[53]
NLS peptides	SV40 T antigen	PKKKRKV	[56,57,58,59,60]
	Mouse FGF3	RLRRDAGGRGGVYEHLGGAPRRRK	[61]
	NLSV404	PKKKRKVGPKKKRKVGPKKKVGPKKKRKVGC	[62]
	Ku7O_2_	CKVTKRKHGAAGAASKRPKGKVTKRKHGAAGAASKRPK	[63]
Other peptides	Smart peptide	Nap-FFGPLGLAG(CK_m_)_n_C	[64]
	24-mer β-annulus peptide	INHVGGTGGAIMAPVAVTRQLVGS	[65,66]
	β-annulus-GGGCG peptide	INHVGGTGGAIMAPVAVTRQLVGSGGGCG	[67]
	H4K5HC_BZl_C_BZl_H	HHHHKKKKKC12LLHC_BZl_C_BZl_HLLGSPD	[68]
	K3C6SPD	KKKC6WLVFFAQQGSPD	[69,70]
	CC	REGVAKALRAVANALHYNASALEEVADALQKVKM	[71]
	Surfactant-like peptide	IIIVVVAAAGGGKKK	[72]

^a^ All peptide sequences are given in the one-letter code amino acid name (Table A1, Appendix A).

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
