# Peer review of "Application of Peptides in Construction of Nonviral Vectors for Gene Delivery"

_nanomaterials, 2022, doi:10.3390/nano12224076_

Round 1

Reviewer 1 Report

Please find the attached PDF document that contains suggestions and comments on the manuscript. 

Reviewer 2 Report

The manuscript entitled “Application of peptides in construction of nonviral vectors for gene delivery” by Yang Y et al. summarized recent progress on the application of peptide for nucleic acid therapeutics delivery by overcoming biological barriers. The manuscript discussed biological barriers and cellular pathways of nucleic acid therapeutics, applications of different categories of peptides for enhanced gene delivery as well as challenges and future perspectives. The manuscript is well structured with clear logic. Following revisions are needed before considered for publishing:

1.       The phrasing is a little problematic in this manuscript; for example: “compensate disease” (Line 30, Page1), “stimuli responsibility” (Line65, Page2), “nonspecific feather” (Line127, Page4). Please do carefully proofread and improve the language.

2.       “The therapeutics genes” used at Line33-34 in Page 1 is not appropriate. Although DNA and siRNA can be used as genetic medicine, they could be called therapeutics agents or nucleic acid therapeutics rather than “genes”. Please correct it through the whole manuscript.

3.       The authors discussed the biological barriers to overcome under introduction and mentioned the vectors should be designed to avoid non-specific interaction with proteins in the blood at Line55 in Page2. However, almost no delivery vehicle can get rid of the interaction with components in the bloodstream. Besides, formulation such as lipid nanoparticles rely on lipoprotein in the blood to deliver therapeutics. The interaction can be “reduced” but not “avoid”.

4.       The structure of the manuscript is oversimplified. This review can be improved by covering more aspects about peptide for gene delivery. It can be considered to add “construction” aspect before the “application” session, such as synthesis of peptide molecules, linker chemistry of peptide-nucleic acid conjugate, methods of making peptide conjugated nonviral vectors, etc.  

5.       It’s good the authors discussed the application of various peptides for delivery by their function with some specific examples in the main text. However, summary and integration are lacking. It can be considered to add a summary table to include function, example peptides, sequence, and references, etc. to provide the overview.

6.       What’s the advantages and disadvantages when comparing direct peptide conjugate to peptide incorporated nonviral vectors to deliver nucleic acid therapeutics? Could authors provide any insights about the potential disease areas to be targeted considering the difference properties of these two delivery approaches?

Round 2

Reviewer 1 Report

 Accepted in present form

Reviewer 2 Report

The authors have addressed all questions properly and I agree the current manuscript to publish in Nanomaterials.